# Divergent Changes in Plasma AGEs and sRAGE Isoforms Following an Overnight Fast in T1DM

**DOI:** 10.3390/nu11020386

**Published:** 2019-02-13

**Authors:** Edwin R. Miranda, Kelly N. Z. Fuller, Ryan K. Perkins, Paul J. Beisswenger, Sarah S. Farabi, Lauretta Quinn, Jacob M. Haus

**Affiliations:** 1School of Kinesiology, University of Michigan, 401 Washtenaw Ave., Ann Arbor, MI 48109, USA; edwinray@umich.edu (E.R.M.); ryperkin@umich.edu (R.K.P.); 2Department of Molecular and Integrative Physiology, Kansas University Medical Center, 3901 Rainbow Blvd. Kansas City, KS 66160, USA; kfuller5@kumc.edu; 3Geisel School of Medicine, Dartmouth College, 1 Rope Ferry Rd., Hanover, NH 03755, USA; pjb@preventagehealthcare.com; 4Endocrine, Metabolism, & Diabetes, Division of Medicine, University of Colorado Anschutz Medical Campus, 13001 E 17th Pl., Aurora, CO 80045, USA; schwarzsar@gmail.com; 5Department of Biobehavioral Health Science, University of Illinois at Chicago, 845 Damen Ave., Chicago, IL 60612, USA; lquinn18@gmail.com

**Keywords:** Diurnal Flux, RAGE, soluble RAGE, methylglyoxal, fasting

## Abstract

Advanced glycation end products (AGEs) promote the development of diabetic complications through activation of their receptor (RAGE). Isoforms of soluble RAGE (sRAGE) sequester AGEs and protect against RAGE-mediated diabetic complications. We investigated the effect of an overnight fast on circulating metabolic substrates, hormones, AGEs, and sRAGE isoforms in 26 individuals with type 1 diabetes (T1DM). Blood was collected from 26 young (18–30 years) T1DM patients on insulin pumps before and after an overnight fast. Circulating AGEs were measured via LC-MS/MS and sRAGE isoforms were analyzed via ELISA. Glucose, insulin, glucagon, and eGFR_cystatin-c_ decreased while cortisol increased following the overnight fast (*p <* 0.05). AGEs (CML, CEL, 3DG-H, MG-H1, and G-H1) decreased (21–58%, *p* < 0.0001) while total sRAGE, cleaved RAGE (cRAGE), and endogenous secretory RAGE (esRAGE) increased (22–24%, *p* < 0.0001) following the overnight fast. The changes in sRAGE isoforms were inversely related to MG-H1 (*rho* = −0.493 to −0.589, *p* < 0.05) and the change in esRAGE was inversely related to the change in G-H1 (*rho* = −0.474, *p* < 0.05). Multiple regression analyses revealed a 1 pg/mL increase in total sRAGE, cRAGE, or esRAGE independently predicted a 0.42–0.52 nmol/L decrease in MG-H1. Short-term energy restriction via an overnight fast resulted in increased sRAGE isoforms and may be protective against AGE accumulation.

## 1. Introduction

Advanced glycation end products (AGEs) are a heterogeneous group of glycotoxins formed exogenously via the Maillard Reaction [1] and endogenously via a number of mechanisms. Under normal physiologic conditions, the endogenous formation of AGEs is slow and the clearance rate is adequate to prevent their accumulation [2]. However, the chronic oxidative stress and hyperglycemia conditions that characterize type 1 (T1DM) and Type 2 diabetes mellitus (T2DM), accelerate AGE formation leading to the accumulation of AGE-modified proteins in tissues and in circulation [3,4,5]. AGE-modified proteins in tissues are subsequently degraded releasing AGE-free adducts into the circulation. With diabetes, post prandial hyperglycemia also promotes the production of the highly reactive α-dicarbonyls which include, methylglyoxal (MG), glyoxal (GO) and 3-deoxyglucosone (3DG) [5,6]. These α-dicarbonyls react primarily with arginine residues to form hydroimidazolone AGE modifications on proteins and also contribute to circulating AGE-free adducts. Other prominently studied AGEs include carboxymethyl lysine (CML) and carboxyethyl lysine (CEL). These glycation modifications form on lysine residues of proteins through a number of mechanisms including the Maillard reaction, lipid peroxidation, and degradation of fructosyl lysine adducts [2]. Accumulation of these and other AGEs is well documented as a major contributor to diabetic complications such as neuropathy, nephropathy, and microvascular disease [7,8,9]. AGEs impose these pathologic effects by directly modulating the structures and functions of individual proteins, cells and tissues as well as through binding to their receptor (RAGE) [2,10,11,12,13,14].

RAGE is a type I transmembrane receptor that belongs to the immunoglobulin super family and was first found via its ability to bind to AGE modified proteins [15,16]. CML has drawn particular interest as a ligand for RAGE however, the affinity of this interaction is on the micromolar scale whereas the affinity of RAGE for MG-H1 modified proteins and MG-H1 free-adducts is on the nanomolar scale, an order of magnitude greater [17,18]. Binding of circulating AGEs to full-length, membrane-bound RAGE initiates a signaling cascade leading to nuclear factor kappa-B (NF-κB) activation. Once activated, NF-κB promotes the transcription of several inflammatory cytokines as well as RAGE itself, propagating a futile cycle [19]. RAGE signaling is further able to contribute to this feed-forward process by promoting oxidative stress through the activation of NADPH oxidase [20]. This oxidative environment further contributes to the de novo production of AGEs which may go on to act as RAGE ligands.

Alternatively, circulating soluble RAGE (sRAGE) isoforms lack the intracellular domain of RAGE which is necessary for initiation of downstream signaling [15]. The sRAGE pool is a heterogeneous group of proteins made up of primarily two isoforms: cleaved RAGE (cRAGE) which is produced via proteolytic cleavage of the RAGE ectodomain via ADAM10 (a disintigrin and metalloproteinase 10), and endogenous secretory RAGE (esRAGE) which is produced via alternative splicing of RAGE pre-mRNA [21,22,23]. sRAGE isoforms are believed to act as competitive inhibitors of RAGE and have been repeatedly demonstrated to be protective against AGE-RAGE mediated vascular dysfunction and insulin resistance [15,24,25].

Previous cross-sectional analyses have demonstrated elevated circulating and urine AGEs in individuals with T1DM compared to age matched controls [4,5,26,27]. However, whether sRAGE is protective against, or simply a marker for complications in individuals with T1DM remains inconclusive. Contrary to the majority of findings in individuals with obesity or T2DM, several studies report elevated circulating sRAGE in T1DM individuals to be predictive of diabetic complications and risk of all-cause and CVD mortality [27,28,29,30,31,32,33]. However, poor kidney function has been demonstrated to increase circulating sRAGE and has been shown to confound some of the positive relationships between sRAGE and CVD [27,30]. On the other hand, some studies have demonstrated lower total sRAGE and esRAGE in T1DM patients compared to age-matched controls and an inverse association between sRAGE isoforms and markers of inflammation [31,32,34].

Previous studies examining AGEs and sRAGE are limited by their cross-sectional design whereas interventional studies examining change in AGEs and sRAGE are lacking. In addition, whether or not changes in sRAGE and AGEs are related to one another is also unclear. Thus, understanding the temporal dynamics of sRAGE isoforms and AGE species is paramount to determining their physiological relevance and their efficacy as disease biomarkers. Of the many perturbations that may influence sRAGE and AGE in vivo, the effects of negative energy balance (i.e., caloric restriction or intermittent fasting) are of particular interest given reports of metabolic protection afforded in RAGE null animals fed a high fat diet [35] and the ability of NAD^+^-dependent deacetylase, Sirtuin 1 (SIRT1), to activate ADAM10 transcription [36]. Further, other studies have demonstrated that ADAM10 activity can be stimulated by G-protein coupled receptor (GPCR) signaling [37]. Glucagon may be a candidate effector of this process given its concentration is increased during low energy states such as fasting and signaling through its GPCR. In addition, our work, and others, have demonstrated periods of caloric restriction are able to increase esRAGE transcription and circulating esRAGE concentration in normal weight and obese subjects, respectively [38,39]

Therefore, the purpose of our study was to comprehensively characterize the change in circulating AGEs, oxidative damage markers and sRAGE isoforms following an overnight fast in individuals with T1DM. We hypothesized that the acute negative energy balance imparted by an overnight fast would increase sRAGE isoforms and decrease circulating AGEs. We also explored the ability of sRAGE isoforms and circulating metabolic regulators to predict the reduction of MG-H1, the most abundance AGE-free adduct, using multivariate regression analysis.

## 2. Materials and Methods

### 2.1. Participants

Young adults (*n* = 26, 24 ± 1 years) previously diagnosed with T1DM were recruited from the greater Chicago metropolitan area. To be eligible, subjects were required to be diagnosed with T1DM for at least 5 years and utilize an insulin pump [40]. Subjects were excluded if they were night- or rotating-shift workers, using medication that altered sleep, were diagnosed with cardiovascular disease, uncontrolled thyroid disease, or reported any diabetic complications. The study was approved by the institutional review board at the University of Illinois at Chicago and all subjects provided written informed consent prior to participation in accord with the Declaration of Helsinki (project IRB approval code: 2013-0030).

### 2.2. Clinical Procedures

These data were generated via post hoc analysis of study samples from a patient population that has been previously described [40]. The original study utilized continuous glucose monitoring for three days prior to and the day of the in-patient stay. During the three days prior to the in-patient stay, subjects were free to carry out their normal activities, exercise and eat ad libitum. This post hoc analysis is focused on samples collected during the overnight in-patient stay. Briefly, participants arrived at the College of Nursing at the University of Illinois at Chicago at 2000 h. Subjects were instructed to consume their regular evening meal; however, the composition of this meal was not controlled. Blood samples collected with EDTA anticoagulant were obtained immediately before lights-out (22:00) and immediately after lights-on (06:00) and centrifuged to isolate plasma, after which plasma was aliquoted and stored at −80 °C for future analysis. Subjects spent at least 7 h in bed and the fasting period was approximately 10 h.

### 2.3. Quantification of Circulating Protein Oxidation, and Glycation Free Adducts via LC-MS/MS

Circulating oxidative damage products methionine sulfoxide (MethSO), 2-Amino adipic acid (2-AAA), and AGE-free adducts MG-H1, G-H1, CML, CEL, and 3DG-H were measured via isotope dilution analysis liquid chromatography-tandem mass spectrometry (LC-MS/MS) with an Agilent model 6410 triple quadrupole MS system with 1200 Rapid Resolution 1200 LC system as described previously [41]. Briefly, oxidative damage markers and AGE-free adducts were quantified in plasma filtrates prepared via centrifugation through 10 K cut-off Amicon^®^ filters and separated by liquid chromatography with a methanol/H_2_O gradient mobile phase with 0.29% heptafluorobutyric acid (HBFA).

### 2.4. Quantification of Circulating sRAGE Isoforms

Circulating total sRAGE (R&D Systems Inc., Minneapolis, MS, USA) and esRAGE (As One International, Mountain View, CA, USA) were determined by commercially available ELISAs per manufacturer’s protocol. Circulating cRAGE was calculated by subtracting esRAGE from total sRAGE as previously described [39,42,43,44]. We also derived a ratio of cRAGE:esRAGE to examine the proportional expression of the isoforms as previously described [39,42,43,44]. Given that cRAGE and esRAGE are likely generated by independent mechanisms, deriving a ratio of the two isoforms gives insight into the relative contribution of the mechanisms.

### 2.5. Quantification of Circulating Metabolic Substrates and Hormones

Plasma non-esterified fatty acids (NEFAs) were determined via colorimetric assay according to manufacturer’s protocol (Wako Pure Chemical Industries Ltd., Osaka, Japan). Total cholesterol, LDL, HDL, Non-HDL, and triglycerides were determined via Cholestech LDX lipid profile cassettes (Alere Inc., Hayward, CA, USA). Glucose was measured using a bed-side analyzer (YSI Stat, Yellow Springs, USA; ABL, Radiometer, Denmark). Insulin and glucagon were measured via commercially available ELISAs according to manufacturer’s protocol (Crystal Chem Inc., Elk Grove Village, IL, USA). Cystatin-C was measured via ELISA (R&D Systems Inc., Minneapolis, MS, USA). Plasma concentrations of Cystatin-C were subsequently used to calculate estimated glomerular filtration rate via the following equation:(1)$${\mathrm{eGFR}}_{\text{Cystatin-C}}\left(\mathrm{mL}\cdot \min^{-1}\cdot 1.73\;m^{-2}\right)\;=\;\left(\;\frac{84.6}{\mathrm{Cystatin}-C\left(\frac{\mathrm{mg}}{L}\right)}\right)-3.2$$

This equation was first described by MacIsaac et al. and was later validated against isotopic measurement of GFR [45,46]. Cortisol, IL-6, and TNF-α were measured via ELISA (R&D Systems Inc., Minneapolis, MS, USA) and have been previously reported along with HbA1c [40,47].

### 2.6. Data Analysis/Statistics

All statistical analyses were performed using SPSS version 24 (IBM, Armonk, NY, USA). Data were tested for normality via Shapiro-Wilk test. Comparisons between time points were made via student’s paired T test or Wilcoxon Sign Rank test where appropriate. Relationships between variables were analyzed via Pearson’s or Spearman’s correlation where appropriate. Multivariate regression analysis was used to determine the effect of the changes in sRAGE isoforms, glucose, eGFR_Cystatin-C_, and glucose counterregulatory hormones (glucagon, insulin, and cortisol) on MG-H1. To avoid bias, independent variables were entered into the models simultaneously rather than using a stepwise model. Sex was not considered in the analysis given our small *n* size. All data are presented as mean ± SEM and differences were deemed significant if *p* < 0.05.

## 3. Results

### 3.1. Baseline and Metabolic Changes Following an Overnight Fast

Baseline anthropometric and metabolic characteristics are presented in Table 1. Two subjects in the cohort were former smokers (> 1 year) and one was a current smoker. Table 1 presents metabolic values before (22:00) and after (06:00) the in-clinic, overnight fast. Following the overnight fast, glucose (−2.83 ± 0.86 mmol/L), insulin (−59.25 ± 10.91 pmol/L), glucagon (−7.15 ± 1.89 ng/L), and IL-6 (−0.18 ± 1.41 pg/mL) decreased (*p* < 0.05, Table 1). Conversely, cortisol (323.6 ± 23.7 nmol/L), and TNF-α (0.47 ± 0.05 pg/mL) increased following the overnight fast (*p* < 0.05).

### 3.2. Divergent Changes Between sRAGE Isoforms, AGEs/Oxidative Stress Markers with Overnight Fast

Total sRAGE (+270 ± 44.4 pg/mL), cRAGE (+206 ± 35.3 pg/mL), and esRAGE (+63.9 ± 10.7 pg/mL) increased following the overnight fast (*p* < 0.001, Figure 1A–C) whereas cRAGE:esRAGE did not change (*p* > 0.05, Figure 1D). Conversely, AGE-free adducts MG-H1 (−189 ± 25.1 nmol/L), G-H1 (−3.32 ± 0.67 nmol/L), CML (−33.8 ± 9.33 nmol/L), CEL (−19.0 ± 3.87 nmol/L), and 3DG-H (−161 ± 35 nmol/L) all decreased following the overnight fast (*p* < 0.0001, Figure 2 C–G). Circulating markers of protein oxidation MethSO (−864 ± 91.2 nmol/L) and 2-AAA (−651 ± 140 nmol/L) also decreased (*p* < 0.0001, Figure 2A,B). The change in esRAGE was negatively correlated to the change in G-H1 (*rho* = −0.474, *p* = 0.02) and the change in all sRAGE isoforms were negatively correlated with the change in MG-H1 (Table 2). Neither the changes in sRAGE isoforms, nor the changes in AGEs, were related to changes in circulating oxidative stress markers.

### 3.3. Diurnal Changes in eGFR_Cystatin-C_ are Related to Changes in MG-H1 but Does Not Affect the Relationships between MG-H1 and sRAGE Isoforms

Given the relationship between the change in MG-H1 and the change in sRAGE, we next determined if diurnal changes in eGFR_Cystatin-C_ were related to changes in MG-H1 or sRAGE isoforms. This was an important consideration since AGE-free adducts are filtered by the kidney and sRAGE may be as well given its Stokes radius is sufficiently small (2.81 nm) [48] to traverse glomerular pores whose size is approximately 8 nm [4,26,41]. As expected, the change in eGFR_Cystatin-C_ negatively correlated with the change in MG-H1 (*rho* = −0.419, *p* = 0.023). We then examined the relationship between the change in sRAGE isoforms and MG-H1 while controlling for the change in eGFR_Cystatin-C_ by utilizing partial correlations. After controlling for the changes in eGFR_Cystatin-C_, the relationships between the change in total sRAGE (*r* = −0.669, *p* < 0.001), cRAGE (*r* = −0.670, *p* < 0.001), and esRAGE (*r* = −0.564, *p* < 0.01) and MG-H1 remained significant suggesting minimal effect of the change in renal function on the interaction between sRAGE isoforms and MG-H1. Further, we explored the possibility that diurnal changes in eGFR_Cystatin-C_, glucose and glucose counterregulatory hormones (insulin, glucagon, and cortisol) influenced the relationship between the changes in sRAGE isoforms and MG-H1. These independent variables were simultaneously entered into linear regression models with the change in MG-H1 as the dependent variable (Table 3). MG-H1 was chosen as the dependent variable given the significant relationships between the change in sRAGE isoforms and the change in MG-H1. In each of the models, the change in total sRAGE, cRAGE, and esRAGE were the only significant contributors to the models although the changes in eGFR_Cystatin-C_ and cortisol were trending toward significance (Table 3). Changes in each of the sRAGE isoforms predicted the change in MG-H1 whereby every 1 pg/mL increase in total sRAGE, cRAGE, and esRAGE predicted a 0.292 nmol/L, 0.364 nmol/L, and 0.972 nmol/L decrease in MG-H1 respectively (unstandardized B of −0.292, −0.364, and −0.927 respectively). Interestingly, the models that utilized the change in total sRAGE (Adjusted R^2^ = 0.587, *p* < 0.01) and cRAGE (R^2^ = 0.587, *p* < 0.01) were the strongest in predicting changes in MG-H1 (Table 3).

## 4. Discussion

Circulating AGEs tend to accumulate in T1DM and predict the development of complications [2,4,5,49], whereas the ability for sRAGE isoforms to do the same have been equivocal. So far, the literature with regard to sRAGE isoforms have mainly been limited to cross-sectional studies and the temporal dynamics of these factors are poorly understood. AGEs directly affect the manifestation of diabetic complications by altering protein structure and function and indirectly through via activation of RAGE signaling. Soluble RAGE acts to sequester AGEs therefore, it is not surprising that administration of sRAGE both in vitro and in vivo attenuates AGE/RAGE-mediated complications such as atherosclerosis [24], and insulin resistance [25]. These data are the first to demonstrate a relationship between a decrease in circulating AGE-free adducts with a concomitant increase in sRAGE isoforms following an overnight fast in T1DM patients. Acknowledging previous work in T1DM demonstrating increased sRAGE as a risk factor for CVD [27,30], our data would appear in conflict to these findings. However, our work was not designed to examine CVD risk but rather, physiological diurnal variations of these biomarkers. Nevertheless, sRAGE isoforms in T1DM individuals are greater than age-matched controls without T1DM [27,28,29,30], although the exact mechanisms for these observations are yet to be elucidated.

Soluble RAGE is produced via two independent mechanisms: cleavage of the RAGE ectodomain by matrix metalloproteinases, such as ADAM10, to produce cRAGE [21] and alternative splicing of the RAGE gene (*Ager*) to produce esRAGE [22]. Transcription and activity of ADAM10 are regulated by the transcription factor PPARα [23] which promotes transcription of genes involved in fat catabolism, and the NAD^+^-dependent deacetylase SIRT1 [36] which modulates autophagy and mitochondria biogenesis signaling. Both of these pathways are activated during periods of low energy availability such as during fasting and may explain the increase in cRAGE we observed following an overnight fast. The production of esRAGE is not well understood but has been reported to be inhibited by the splicing silencer heterogeneous nuclear RNA binding protein A1 (hnRNPA1) and promoted by the splicing enhancer transformer 2β (Tra2β) in neuronal cells [50]. Regulation of Tra2β may also be related to energy status as individuals with obesity have lower skeletal muscle and adipose expression of Tra2β compared to lean individuals [51]. In support of the ability of low energy state to promote sRAGE production, our lab previously demonstrated increases in esRAGE following a 24-week weight-loss intervention utilizing alternate day fasting as a dietary strategy [39] and conversely lower circulating esRAGE in individuals with obesity compared to lean individuals [43].While the current study is limited by the lack of cells or tissue samples to examine abundance of membrane-bound/full-length RAGE, or the mechanisms of sRAGE production, our data suggest that the fasting state may be able to provoke sRAGE production by either the aforementioned mechanisms, or through mechanisms yet to be elucidated. Certainly, we cannot conclude that fasting has direct mechanistic influence on reducing membrane-bound/full-length RAGE expression, or RAGE-mediated signaling. Future studies should have concomitant measures of cellular or tissue RAGE expression, intracellular signaling, circulating AGEs, and sRAGE isoforms to address these nebulous areas.

Our observed correlation between changes in sRAGE isoforms and changes in MG-H1 free adducts suggest that sRAGE isoforms may be able to sequester MG-H1. However, it is not clear why similar relationships with other AGE-free adducts were not observed. RAGE affinity for MG-H1 is on the nmol/L scale [18], whereas other AGEs (e.g., CML), are on the μM scale [52]. Further, MG-H1 was the most abundant AGE-free adduct in our cohort and has been reported to be approximately ten-fold higher in T1DM compared to age-matched healthy individuals [4]. The ability of sRAGE to sequester circulating AGEs in a physiologically meaningful way is often criticized because of the large concentration difference between the decoy and the ligand. In the current study, we calculated a MG-H1 to total sRAGE ratio of 60 ± 5.7 (Mean ± SEM) at 2200 h which was reduced to 19 ± 1.8 at 06:00 (*p* < 0.001). This suggests a potential indirect mechanism by which sRAGE is able to attenuate circulating MG-H1 concentrations other than simply sequestration of the adduct.

The main clearance mechanism of AGEs is believed to be through filtration by the kidney [26]. In addition, both AGEs and sRAGE have been previously shown to be inversely related to renal function [4,7,27]. In accord with previous literature, we observed a negative correlation between baseline eGFR_Cystatin-C_ and CML (*rho* = −0.420, *p* < 0.05), despite all of our participants having normal kidney function. However, we did not observe any correlations between eGFR_Cystatin-C_ and sRAGE isoforms at baseline. Compared to existing literature describing a relationship between renal function and sRAGE, our cohort of T1DM was younger [27] which may explain why we were not able to recapitulate this result. We also demonstrate, via multiple regression analysis, that eGFR_Cystatin C_ does not contribute to any of the models predicting the change in MG-H1. However, while the change in eGFR was statistically significant, the decrease in eGFR observed in this context is not clinically relevant. With these factors in mind, it is not surprising that eGFR did not contribute to predicting the change in MG-H1 free adducts although the trending *p* value suggests that a more robust stimulus/intervention, with a larger *n* size or a more pathological group of T1DM patients may implicate renal function to have a role in this relationship.

Other factors that have been demonstrated to alter AGE and sRAGE are glucose and insulin. Several investigations in individuals with T1DM and T2DM have demonstrated that elevated plasma glucose, during an oral glucose tolerance test, or by administering a mixed meal increases circulating reactive dicarbonyls which may lead to increased circulating AGEs. Indeed, in our cohort, the change in glucose was positively correlated to the change in MG-H1 free adducts (*r* = 0.379, *p* = 0.037) [5,6,53]. In addition, insulin has been previously shown to promote sRAGE production and has been suggested as a potential explanation for many studies demonstrating elevated sRAGE values in T1DM [54]. Given these data, we included the major glucose counterregulatory hormones and the change in glucose in our regression models. However, neither the change in glucose nor did any of the glucose counterregulatory hormones significantly contribute to predicting the change in MG-H1. Therefore, the regression models demonstrated that the change in sRAGE isoforms independently accounted for more than 50% of the variability in the change in MG-H1 following an overnight fast.

More work is needed to determine the mechanisms that explain this relationship between sRAGE isoforms and MG-H1. Perhaps sRAGE isoforms are indeed able to sequester and remove sufficient amounts of MG-H1 from the circulation. Another possibility is that sRAGE has a more indirect effect on MG-H1 adduct appearance through modulating cellular receptors and downstream inflammation which has been previously suggested of sRAGE [55,56]. Importantly, few studies simultaneously report both sRAGE isoforms and AGE-free adducts, many of which are of cross-sectional design and rely on skin autofluorescence as a surrogate AGE marker [28,57]. A distinguishing characteristic of our work is the concurrent reporting of AGEs and sRAGE isoforms and that we examined both before and after a physiological perturbation with known cardiometabolic mechanisms at play.

Interpretation of these data should be done with caution given our limited sample size and absence of a control group without T1DM. We also did not collect urine or tissue samples and are thus limited to speculate on the tissue specific mechanisms of sRAGE production with fasting or clearance of sRAGE and AGE-free adducts. These points should be a major focus of future studies to determine if targeting sRAGE-producing mechanisms is viable for attenuating AGE burden and, if doing so confers positive health outcomes in individuals with diabetes.

As mentioned previously, two of our participants were former smokers and one was a current smoker at the time of study. There is an established influence of smoking on the concentrations of both sRAGE isoforms and circulating AGEs, which was recently reviewed by Prasad et al. [58]. However, the changes in sRAGE isoforms and AGEs observed in the entire cohort were mirrored in these individuals, and removing these individuals from the analyses did not alter any of the outcomes. A final consideration when interpreting these data is that our baseline measures were made 3–4 h postprandial and we did not standardize the participants’ dietary compositions. The AGE composition of a typical western diet has been shown to influence various metrics of metabolism [59,60,61,62]. Indeed, we recently demonstrated the importance of dietary composition on sRAGE where we found that a high fat meal decreases circulating sRAGE concentrations with concomitant increases in blood mononuclear cell RAGE protein expression of lean healthy individuals [44]. Nevertheless, it is important to note that much of our daylight hours are spent in the post-prandial state and the total AGE burden, independent of source (exogenous versus endogenous) is a driver of inflammation and insulin resistance. Thus, elucidation of strategies and perturbations that elicit divergent changes is sRAGE isoforms and AGE-free adducts, such as a period of fasting, has important implications for cardiometabolic health and in the future design of studies pertaining to AGE-RAGE biology.

## 5. Conclusions

In conclusion, the data presented herein demonstrate the ability of fasting to increase sRAGE isoforms and that these changes are strongly related to decreases in circulating MG-H1 adducts with fasting. These data provide further evidence for the potential therapeutic effect of sRAGE on preventing and treating diabetes and its complications.

## Figures and Tables

**Figure 1 nutrients-11-00386-f001:**
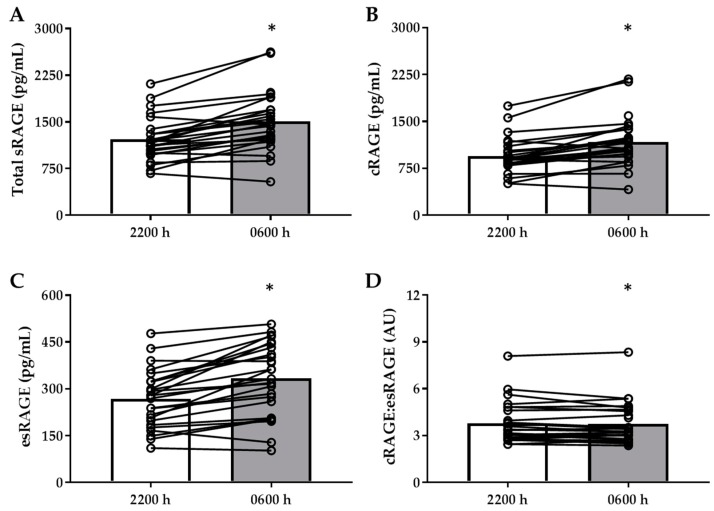
Plasma concentrations of soluble activation of their receptor (sRAGE) isoforms following an overnight fast. (**A**) Plasma concentration of total sRAGE, (**B**) plasma concentration of cleaved sRAGE (cRAGE), (**C**) plasma concentration of endogenous secretory RAGE (esRAGE), (**D**) proportion of plasma cRAGE:esRAGE before and after an overnight fast in T1DM subjects. Data were analyzed via paired t-test and are presented as mean with individual data plotted. * Indicates significance difference between time points (*p* < 0.0001).

**Figure 2 nutrients-11-00386-f002:**
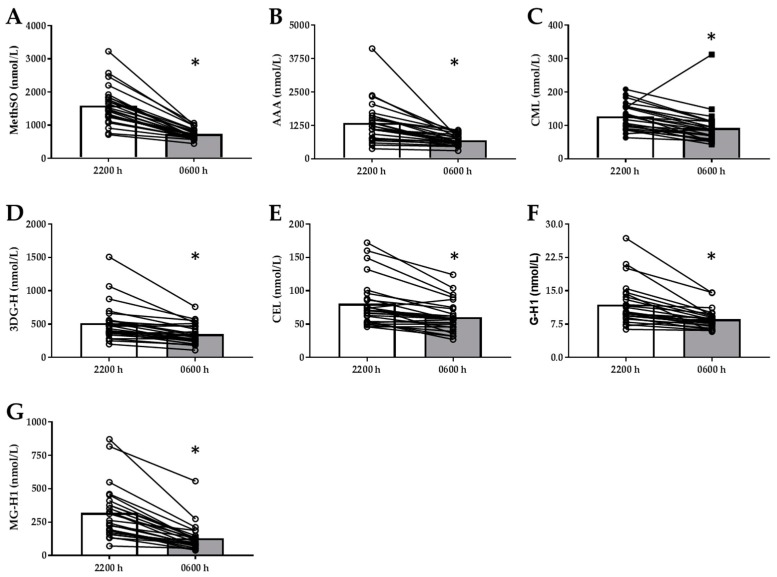
Plasma concentrations of circulating oxidation markers and age-free adducts following an overnight fast. (**A**) Plasma concentration of methionine sulfoxide (MethSO), (**B**) plasma concentration of amino adipic acid (AAA) (**C**) Plasma concentration of carboxymethyl lysine (CML), (**D**) plasma concentration of 3-Deoxyglucosone Hydroimidazalone (3DG-H), (**E**) carboxyethyl lysine (CEL), (**F**) glyoxal hydroimidazolone (G-H1) and (**G**) methylglyoxal hydroimidazalone (MG-H1) free adducts before and after an overnight fast in T1DM subjects. Data were analyzed via paired t-test and are presented as mean with individual data plotted. * Indicates significance difference between time points (*p* < 0.0001).

**Table 1 nutrients-11-00386-t001:** Anthropometric, metabolic and hormonal measures.

	22:00	06:00	*p*
**Gender (M/F)**	14/12		N/A
**Age (years)**	24 ± 1		N/A
**Diabetes Duration (y)**	12 ± 1		N/A
**Weight (kg)**	75.6 ± 2.5		N/A
**BMI (kg/m^2^)**	26.2 ± 0.7		N/A
**HbA1C (%)**	7.69 ± 0.27		N/A
**Avg Overnight Glucose (mmol/L)**	7.27 ± 0.58		N/A
**Glucose (mmol/L)**	8.46 ± 0.86	5.67 ± 0.59	**0.003**
**eGFR_cystatin-C_ (mL ·min^−1^ ·1.73^−2^)**	109.7 ± 2.8	107.2 ± 5.2	**0.011**
**Insulin (pmol/L)**	113.4 ± 12.9	54.1 ± 3.5	**<0.001**
**NEFA (mEq/L)**	0.32 ± 0.04	0.34 ± 0.04	0.459
**Total Cholesterol (mmol/L)**	4.65 ± 0.18	4.63 ± 0.20	0.621
**Triglycerides (mmol/L)**	1.10 ± 0.11	1.02 ± 0.12	0.073
**HDL (mmol/L)**	1.51 ± 0.09	1.42 ± 0.08	0.157
**LDL (mmol/L)**	2.52 ± 0.16	2.68 ± 0.16	0.397
**Non-HDL (mmol/L)**	3.03 ± 0.18	3.04 ± 0.18	0.791
**IL-6 (pg/mL)**	1.02 ± 0.11	0.83 ± 0.11	**0.020**
**TNF-α (pg/mL)**	0.95 ± 0.09	1.43 ± 0.13	**<0.001**
**Cortisol (nmol/L)**	822.9 ± 3.8	1712.3 ± 24.7	**<0.001**
**Glucagon (ng/L)**	15.4 ± 2.4	8.9 ± 1.2	**0.002**

Data are represented as Mean ± SEM. Abbreviations: BMI, body mass index, HbA1C, glycated hemoglobin protein A1C, eGFR_cystatin-C_, estimated glomerular filtration rate, NEFA, non-esterified fatty acids, HDL, high density lipoprotein, LDL, low density lipoprotein, IL-6, interleukin-6, TNF-α, tumor necrosis factor alpha. Bolded *p* values indicate significant change in parameter.

**Table 2 nutrients-11-00386-t002:** Relationships between the change in srage isoforms and age-free adducts following an overnight fast.

	∆Total sRAGE (pg/mL)		∆cRAGE (pg/mL)		∆esRAGE (pg/mL)		∆cRAGE:esRAGE	
	Corr.	*p*	Corr.	*p*	Corr.	*p*	Corr.	*p*
**∆MethSO (nmol/L)**	−0.007	0.973	−0.006	0.978	−0.011	0.960	−0.105	0.617
**∆AAA (nmol/L)**	0.121	0.565	0.088	0.674	0.124	0.556	−0.153	0.465
**∆CML (nmol/L)**	−0.112	0.594	−0.162	0.438	−0.071	0.737	−0.169	0.419
**∆3DG−H (nmol/L)**	−0.041	0.847	−0.084	0.690	0.121	0.565	−0.353	0.083
**∆CEL (nmol/L)**	−0.235	0.259	−0.274	0.185	−0.195	0.351	−0.145	0.489
**∆G−H1 (nmol/L)**	−0.295	0.153	−0.274	0.185	**−0.474**	**0.017**	0.112	0.596
**∆ MG −H1 (nmol/L)**	**− 0.505**	**0.010**	**−0.493**	**0.012**	**− 0.589**	**0.002**	0.103	0.624

Relationships were analyzed via Pearson’s R or Spearman’s Rho where appropriate. Significant relationships are bolded (*p* < 0.05).

**Table 3 nutrients-11-00386-t003:** Change in sRAGE isoforms independently predict change in mg-h1 via multiple linear regression models.

Dependent Variable: ΔMG-H1 (nmol/L)				
	Adjusted R^2^	Standardized β	95% CI	*p* Value
**Model 1**	**0.587**	**-**	**-**	**0.0003**
**ΔTotal sRAGE (pg/mL)**	**-**	**−0.517**	**−0.492, −0.093**	**0.007**
**Δ eGFR_Cystatin-C_ (mL ·min^−1^ ·1.73^−2^)**	-	−0.294	−7.43, 0.202	0.062
**ΔGlucose (mmol/L)**	-	0.242	−1.50, 0.940	0.144
**ΔGlucagon (ng/L)**	-	−0.083	−6.23, 3.854	0.620
**ΔInsulin (pmol/L)**	-	0.132	−3.20, 7.57	0.402
**ΔCortisol (nmol/L)**	-	−0.288	−1.88, 0.105	0.076
**Model 2**	**0.587**	**-**	**-**	**0.002**
**ΔcRAGE (pg/mL)**	**-**	**−0.511**	**−0.613, −0.116**	**0.007**
**Δ eGFR_Cystatin-C_ (mL ·min^−1^ ·1.73^−2^)**	-	−0.309	−7.59, 0.006	0.050
**ΔGlucose (mmol/L)**	-	0.221	−0.187, 0.909	0.182
**ΔGlucagon (ng/L)**	-	−0.090	−6.33, 3.74	0.593
**ΔInsulin (pmol/L)**	-	0.138	−3.01, 7.64	0.380
**ΔCortisol (nmol/L)**	-	−0.302	−1.91, 0.056	0.063
**Model 3**	**0.502**	**-**	**-**	**0.006**
**ΔesRAGE (pg/mL)**	**-**	**−0.415**	**−1.87, −0.073**	**0.036**
**Δ eGFR_Cystatin-C_ (mL·min^−^^1^ ·1.73^−2^)**	-	−0.284	−7.74, 0.770	0.102
**ΔGlucose (mmol/L)**	-	0.318	−0.75, 1.11	0.083
**ΔGlucagon (ng/L)**	-	−0.143	−7.53, 3.39	0.433
**ΔInsulin (pmol/L)**	-	0.186	−2.75, 8.89	0.279
**ΔCortisol (nmol/L)**	-	−0.286	−1.98, 0.226	0.111

Bold text indicates significant relationship by regression analysis.

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
