# Peer review of "Divergent Changes in Plasma AGEs and sRAGE Isoforms Following an Overnight Fast in T1DM"

_nutrients, 2019, doi:10.3390/nu11020386_

Round 1

Reviewer 1 Report

Miranda et al. have an interesting paper on the Divergent Changes in Plasma AGEs and sRAGE 2 Isoforms Following an Overnight Fast in T1DM.

It is very well-written. No significant flaws, I just have two comments:

1. The authors mentioned in the Results section line 171 that one subject was a current smoker, and two were former smokers. According to several reports smokers have a higher concentration of some AGEs, and it could also influence on sRAGE concentration*. Therefore, the authors should address this issue in the discussion section.

* Prasad K, Dhar I, Caspar-Bell G. Role of Advanced Glycation End Products and Its Receptors in the Pathogenesis of Cigarette Smoke-Induced Cardiovascular Disease. Int J Angiol. 2015 Jun;24(2):75-80.

2. In page 7, line 231 and 232 the authors mention that

In page 7, Table 3 the standardized beta for ΔcRAGE and ΔesRAGE are -0,511 and -0.415. The authors should review line 231 and 232 where they stated

every 1 pg/mL increase in total sRAGE, cRAGE, and esRAGE predicted a 0.517 nmol/L, 0.511 nmol/L, and 0.415 nmol/L decrease in MG-H1 respectively.

Usually, the standardized beta is lower than the B unstandardized coefficient; therefore the values 0.511 and 0.415 should be reviewed.

Author Response

Reviewer 1

Miranda et al. have an interesting paper on the Divergent Changes in Plasma AGEs and sRAGE Isoforms Following an Overnight Fast in T1DM.

It is very well-written. No significant flaws, I just have two comments:

1. The authors mentioned in the Results section line 171 that one subject was a current smoker, and two were former smokers. According to several reports, smokers have a higher concentration of some AGEs, and it could also influence on sRAGE concentration*. Therefore, the authors should address this issue in the discussion section.

* Prasad K, Dhar I, Caspar-Bell G. Role of Advanced Glycation End Products and Its Receptors in the Pathogenesis of Cigarette Smoke-Induced Cardiovascular Disease. Int J Angiol. 2015 Jun;24(2):75-80.

RESPONSE: We thank the reviewer for highlighting the interaction between smoking and circulating sRAGE and AGE concentrations. We agree and have added information regarding this caveat to the discussion as requested.

2. In page 7, line 231 and 232 the authors mention that:

In page 7, Table 3 the standardized beta for ΔcRAGE and ΔesRAGE are -0,511 and -0.415. The authors should review line 231 and 232 where they stated:

every 1 pg/mL increase in total sRAGE, cRAGE, and esRAGE predicted a 0.517 nmol/L, 0.511 nmol/L, and 0.415 nmol/L decrease in MG-H1 respectively. 

Usually, the standardized beta is lower than the B unstandardized coefficient; therefore, the values 0.511 and 0.415 should be reviewed.

RESPONSE: Thank you for your careful review of the data. Upon reexamination of the data, we realized these mistakes and have made the following corrections. The unstandardized β values for Δ total sRAGE, Δ cRAGE, and Δ esRAGE are -0.292, -0.364, and -0.972 nmol/L respectively. These are now referenced in the results when discussing the change in sRAGE isoforms’ ability to predict changes in MG-H1. We apologize for this oversight.

In addition, upon rechecking the statistics, we realized that the standardized β value for the Δ total sRAGE was misrepresented in the table as -0.397 where it is truly -0.517. The standardized β values for Δ total sRAGE, Δ cRAGE, and ΔesRAGE are indeed -0.517, -0.511, and -0.415 nmol/L which is now represented in table 3.

Reviewer 2 Report

This is a novel study with a reasonably good design and  discussion of results.

1. The authors claim a relationship between decrease in AGE adducts and sRAGE levels in patients with T1DM. While their point is valid, however, it is lacking data on cellular RAGE. Given that the group has previously demonstrated ability to analyze cellular RAGE,  the study would have been strengthened if they could show a change or even no change in membrane bound RAGE on mononuclear cells in blood if tissue sampling was not feasible.

2. While the authors acknowledge the limitations of the study, the conclusion is not supported by any mechanistic evidence and therefore, needs to be revised and limited based on results of the study.  

Author Response

Reviewer 2

This is a novel study with a reasonably good design and discussion of results.

1. The authors claim a relationship between decrease in AGE adducts and sRAGE levels in patients with T1DM. While their point is valid, however, it is lacking data on cellular RAGE. Given that the group has previously demonstrated ability to analyze cellular RAGE, the study would have been strengthened if they could show a change or even no change in membrane bound RAGE on mononuclear cells in blood if tissue sampling was not feasible.

RESPONSE: We appreciate the reviewer’s interest in our work and thank them for their careful review. We agree that a measurement of cellular/tissue RAGE would lend valuable insight into RAGE/sRAGE regulation and strengthen the current study. However, this was a retrospective analysis of frozen samples where the original study design was not developed with these outcomes in mind. The reviewer is correct that we have made these types of measures in the past, however no cells or tissue are available for the current study.  Certainly, future studies and efforts will be made to encompass these variables and to provide additional mechanistic insight.  We have adjusted the discussion to soften our conclusions in light of these comments and comments from Reviewer 3.

2. While the authors acknowledge the limitations of the study, the conclusion is not supported by any mechanistic evidence and therefore, needs to be revised and limited based on results of the study.  

RESPONSE: In an attempt to reiterate the correlative nature of our findings, we have modified the concluding statement as follows:

In conclusion, the data presented here demonstrate the ability of fasting to increase sRAGE isoforms and that these changes are strongly related to decreases in circulating MG-H1 adducts with fasting. These data provide further evidence for the potential therapeutic effect of sRAGE on preventing and treating diabetes and its complications.”

Reviewer 3 Report

This is a well performed study with very interesting results. I have learned quite a bit from reviewing this manuscript.

Issues:

1)    Unfortunately, the study measured only circulating sRAGE and its relationship to simultaneous circulating AGE levels, but not the full cellular RAGE and therefore some of the conclusions are limited. We know from previous studies that sRAGE levels have a direct association with sAGEs and cardiovascular complications in diabetic patients (ref 37), findings that place doubt about the concept that sRAGE acts as a protective binder against circulating AGEs. Of interest, previous work done by the current authors (ref 39) shows that an acute fat meal load (rich in AGEs, by the way) leads to increase cellular RAGE, while sRAGE decreases. This study (ref 39) was in a way the reversal of the current one although here we do not know the food ingested prior to overnight fasting and the findings may fit although we do not have full RAGE here and refer 39 did not have sAGEs. The main point here being that we cannot conclude much about the  sAGE-RAGE axis in the absence of measurement of the cellular RAGE responsible for initiating the cellular events in response to AGEs. The current discussion should moderate itself on the basis of these previous findings.

2)    The decrease in circulating MG levels may be in part due to the simultaneous fall in blood glucose. The acute (less than 10 hours) changes in AGEs such as CML and CEL may also reflect changes glucose-induced MG metabolism, but likely also reflect the in oral supply of such AGEs during fasting. This point should be addressed more clearly in the discussion.

3)    Although I agree that changes in GFR were statistically significant, from a clinical perspective a delta GFR of 2.5 from a baseline of 109.7 ml/min is of no significance. Therefore all the statistical calculations using changes in GFR only increase the noise of data, but they are not really valuable. The authors may leave of this discussion but emphasized the clinical unimportance of the changes observed in GFR.

Author Response

Reviewer 3

This is a well performed study with very interesting results. I have learned quite a bit from reviewing this manuscript.

Issues:

1)    Unfortunately, the study measured only circulating sRAGE and its relationship to simultaneous circulating AGE levels, but not the full cellular RAGE and therefore some of the conclusions are limited. We know from previous studies that sRAGE levels have a direct association with sAGEs and cardiovascular complications in diabetic patients (ref 37), findings that place doubt about the concept that sRAGE acts as a protective binder against circulating AGEs. Of interest, previous work done by the current authors (ref 39) shows that an acute fat meal load (rich in AGEs, by the way) leads to increase cellular RAGE, while sRAGE decreases. This study (ref 39) was in a way the reversal of the current one although here we do not know the food ingested prior to overnight fasting and the findings may fit although we do not have full RAGE here and refer 39 did not have sAGEs. The main point here being that we cannot conclude much about the  sAGE-RAGE axis in the absence of measurement of the cellular RAGE responsible for initiating the cellular events in response to AGEs. The current discussion should moderate itself on the basis of these previous findings.

RESPONSE: We thank the reviewer for their thorough evaluation of our work and we are glad that it was insightful for the reviewer. We agree that a measure of cellular/tissue RAGE would lend valuable insight into the AGE/RAGE axis that is not offered in the current manuscript. Unfortunately, we were limited by the samples available to us and therefore were unable to make a meaningful measure of membrane-bound RAGE. This point was also highlighted by Reviewer 2.  Based upon these comments and comments from Reviewer 2, we have modified our conclusions to be more reflective of our non-mechanistic observations.

2)    The decrease in circulating MG levels may be in part due to the simultaneous fall in blood glucose. The acute (less than 10 hours) changes in AGEs such as CML and CEL may also reflect changes glucose-induced MG metabolism, but likely also reflect the in oral supply of such AGEs during fasting. This point should be addressed more clearly in the discussion.

RESPONSE: We agree with the reviewer’s hypothesis and we tried to address this potential explanation by entering blood glucose into our regression model. However, while the change in glucose was significantly correlated with the change in MG-H1 (r = 0.379, p =0.037) it did not reach significance in the multivariate model. We have added this point to the discussion as requested.

3)    Although I agree that changes in GFR were statistically significant, from a clinical perspective a delta GFR of 2.5 from a baseline of 109.7 ml/min is of no significance. Therefore, all the statistical calculations using changes in GFR only increase the noise of data, but they are not really valuable. The authors may leave of this discussion but emphasized the clinical unimportance of the changes observed in GFR.

RESPONSE: We agree with the reviewer’s comment and have added language to the discussion as requested